# Effect of Optimized Chitosan Coating Obtained by Lactic Fermentation Chemical Treatment of Shrimp Waste on the Post-Harvest Behavior of Fresh-Cut Papaya (*Carica papaya* L.)

Luis Angel Cabanillas-Bojórquez [1,2], Julio Montes-Ávila [3,4], Misael Odín Vega-García [2,4], Héctor Samuel López-Moreno [3,4], Ramón Ignacio Castillo-López [5,*] and Roberto Gutiérrez-Dorado [2,4,*]

1 CONACYT-Centro de Investigación en Alimentación y Desarrollo, A.C., Carretera a Eldorado km 5.5, Col. Campo El Diez, Culiacán CP 80110, Mexico
2 Ciencia y Tecnología de Alimentos, Facultad de Ciencias Químico-Biológicas, Universidad Autónoma de Sinaloa, Ciudad Universitaria, Culiacán CP 80013, Mexico
3 Ciencias Biomédicas, Facultad de Ciencias Químico-Biológicas, Universidad Autónoma de Sinaloa, Ciudad Universitaria, Culiacán CP 80013, Mexico
4 Programa de posgrado en Biotecnología, Facultad de Ciencias Químico-Biológicas, Universidad Autónoma de Sinaloa, Ciudad Universitaria, Culiacán CP 80013, Mexico
5 Ingeniería Química, Laboratorio de Tecnología Poscosecha y Fermentaciones, Facultad de Ciencias Químico-Biológicas, Universidad Autónoma de Sinaloa, Ciudad Universitaria, Culiacán CP 80013, Mexico
* Correspondence: ricastil@uas.edu.mx (R.I.C.-L.); rgutierrez@uas.edu.mx (R.G.-D.)

**Abstract:** Chitosan is a biopolymer obtained from shrimp waste mainly by a polluting chemical method. In this work, a less polluting biological-chemical method to obtain chitosan from this waste has been optimized; this method used a successive lactic fermentation and chemical process. Additionally, in this work, the effect of chitosan coating on the post-harvest behavior of fresh-cut papaya was studied as a practical application. A rotatable central composite design (CCRD) with two variables (fermentation time and total soluble solids of the fermentation medium) was used to optimize the chitosan extraction. The optimized conditions for chitosan extraction were 108 h and 8.74 °Brix. The optimized chitosan showed a high deacetylation degree of 83%, acceptable process yield of 2.03%, a low ash content of 0.23% and a molecular weight of 107.5 kDa. In addition, optimized chitosan decreased the loss of color and acidity, as well as the growth of microorganisms; it also increased the pH of minimally processed papaya slices without a statistically significant difference with that of commercial chitosan. Based on these results, optimized chitosan could be applied to other fruits as a coating to maintain their quality characteristics and inhibit microbial growth during the storage of fresh-cut fruits.

**Keywords:** chitosan coating; lactic fermentation; shrimp waste; optimization; post-harvest papaya

## 1. Introduction

The popularity of marine-based food has been growing continuously, and shrimp is the most economically important one [1]. Shrimp is considered to be a functional food due to its bioactive compounds (proteins, carotenoids and antioxidants) [2], and also, its by-products are a significant source of chitin [3]. Chitin is the second most abundant biopolymer, with a linear chain of β-1,4-linked *N*-acetyl-d-glucosamine. Chitin extraction from crustacean shells is carried out in successive stages: deproteinization, demineralization and the removal of lipids and pigments [4,5]. However, chitin production uses strong alkalis and acids, which cause ecological damage. Additionally, the aggressive process conditions promote depolymerization and produce low-quality chitin. In addition, the chemical treatment for chitin production causes protein denaturation, so it cannot be used in the food industry.

On the other hand, lactic fermentation is a promising process that has been studied to recover chitin and other bioactive compounds, such as protein, lipids and pigments, from shrimp waste, which could be used in the food industry [6]. Bioactive compounds extraction from the lactic fermentation of shrimp waste is less damaging to the environment than chemical extraction is; however, it is still economically unviable. Using lactic acid bacteria from underutilized material for chitin extraction from shrimp waste could replace the expensive and non-environmentally friendly traditional process [7,8].

Chitosan, a natural polymer derived from chitin, which has some applications in the food industry (food packaging and edible coatings), can be affected in terms of its molecular weight and physicochemical properties by its source and method of preparation [9–12]. Chitosan coatings have been used for maintaining the quality characteristics of processing fruits by providing a physical barrier, low oxygen permeability and the antimicrobial activity of chitosan [4,13,14]. In this sense, chitosan coatings have been applied to fresh-cut fruits such as apples, guavas, mangoes, nectarines and papayas [13,15–18].

Papaya (*Carica papaya* L.) is a tropical fruit with high population acceptability and a high content of bioactive compounds. In recent years, researchers have used papaya as a model for the fresh-cut fruit market to hinder their management and extend market presence [13,19]. However, when minimally processed fruits are peeled, sliced and the seeds are removed, it increases the susceptibility of microbial growth and mechanical and physiological losses [13]. In this sense, chitosan coatings have been proven as a good strategy to inhibit microorganism growth and maintain the fresh-cut quality [16,17].

So far, in our literature review, chitosan has been obtained from chitin produced by lactic fermentation or chemical treatments; however, only a few studies have optimized the chitosan production from chitin obtained by successive lactic fermentation chemical processes of shrimp waste. Therefore, this work aims to study the effect of chitosan coatings obtained from the optimized successive lactic acid fermentation chemical treatment of shrimp by-products on maintaining the quality parameters and microorganism inhibition of fresh-cut papaya.

## 2. Materials and Methods

### 2.1. Reagents and Chemicals

Chitosan (a food-grade, odorless and tasteless powder extracted from recycled crab and shrimp shells with 89% deacetylation) was used and supplied by Agrinos AS, Sonora, Mexico. Brain heart infusion agar (Sigma 70138), plate count agar (Sigma 88588) Sabouraud agar (Sigma 84088) were also used. Sodium hydroxide, hydrochloric acid, sodium hypochlorite and the rest of the reagents mentioned in the manuscript were purchased from Sigma-Aldrich (St. Louis, MO, USA).

### 2.2. Biological Material

Shrimp heads (*Litopenaeus vannamei*) were obtained from a local market in Culiacan, Sinaloa, Mexico. Shrimp head preparation was performed according to Cabanillas-Bojórquez et al. [20]. The shrimp waste was washed and crushed before use. Molasses was provided by a local company of sugar and stored at room temperature until use. A commercial mixture of lactic bacteria was inoculated with 3.785 L of commercial milk, and the whey was separated and stored at 25 °C.

### 2.3. Lactic Fermentation

The lactic fermentation process was carried out as previously described by Cabanillas-Bojórquez et al. [20]. Shrimp waste (400 g) was mixed with molasses and whey at a ratio of 1:5 *w/v* to solids at a final volume of 2 L. The lactic fermentation was carried out in a Batch reactor (BioFlo 120 Eppendorf, HH. DE) at 20 °C in anaerobic conditions. The fermentation time and total soluble solids of fermentation medium (°Brix) were studied according to the experimental design (Table 1).

**Table 1.** Experimental design for chitosan optimization from lactic fermentation of shrimp waste.

| Treatment [a] | Process Variables [b] | | | | Response Variables [c] | | |
| | Coded | | Originals | | | | |
| No. [b] | $X_1$ | $X_2$ | FT (h) | SST (°Brix) | $Y_A$ | $Y_Y$ | $Y_{DD}$ |
|---|---|---|---|---|---|---|---|
| 1 | −1 | −1 | 66 | 13 | 0.43 | 1.98 | 90.62 |
| 2 | 1 | −1 | 264 | 13 | 0.24 | 1.80 | 76.10 |
| 3 | −1 | 1 | 66 | 35 | 0.36 | 1.95 | 78.47 |
| 4 | 1 | 1 | 264 | 35 | 0.38 | 1.82 | 79.88 |
| 5 | −1.414 | 0 | 25 | 24 | 0.61 | 1.96 | 87.04 |
| 6 | 1.414 | 0 | 305.01 | 24 | 0.45 | 1.67 | 76.91 |
| 7 | 0 | −1.414 | 165 | 8.44 | 0.18 | 1.94 | 80.38 |
| 8 | 0 | −1.414 | 165 | 39.56 | 0.29 | 2.02 | 77.07 |
| 9 | 0 | 0 | 165 | 24 | 0.31 | 1.65 | 85.57 |
| 10 | 0 | 0 | 165 | 24 | 0.35 | 1.66 | 85.25 |
| 11 | 0 | 0 | 165 | 24 | 0.33 | 1.65 | 86.32 |
| 12 | 0 | 0 | 165 | 24 | 0.31 | 1.61 | 86.34 |
| 13 | 0 | 0 | 165 | 24 | 0.30 | 1.66 | 85.61 |

Rotatable central composite design with two factors and 13 treatments. [a] Does not correspond to the order of processing. [b] $X_1$ = FT = Fermentation time (h); $X_2$ = TSS = Total soluble solids (°Brix). [c] $Y_A$ = Chitosan ash; $Y_Y$ = Process yield; $Y_{DD}$ = Deacetylation degree.

### 2.4. Chitin Extraction

After fermentation, the fermented solid was washed with cold water to remove all the residue solution. Then, chitin extraction was performed following the methodology of Tokatli and Demirdoven [21] with slight modifications. First, a deproteinization stage was performed with a 1M sodium hydroxide solution at a ratio of 1/10 (*w/v*) for 1 h. Next, a demineralization step was performed with a solution of 1N hydrochloric acid at ratio a of 1/10 (*w/v*) for a period of 1 h, and a depigmentation step was performed with a sodium hypochlorite solution 10% under the same conditions described above. At the end of this period, the solid was washed with water until it reached a pH of 7.0, and the solid was dried at 65 °C for 24 h.

### 2.5. Chitosan Production

The production of chitosan was carried out according to Kaya et al. [22] with slight modifications. First, dried chitin was refluxed with 50% NaOH at 115 °C for 4 h at a ratio of 1/15 (*w/v*). Then, the sample was washed with distilled water to a neutral pH and filtered. Finally, the solid was dried at 65 °C for 24 h.

### 2.6. Deacetylation Degree of Chitosan

The degree of deacetylation of chitosan was performed following the methodology of Colina et al. [23] with slight modifications. Dried chitosan was analyzed using an IR spectrometer (Agilent Cary spectrum 600, Agilent Technologies, Santa Clara, CA, USA). Two hundred and fifty milligrams of chitosan were dissolved with 50 mL of acetic acid at 6%; the mixture was placed in plastic molds with a 13 cm diameter, and then dried at room temperature for 24 h in the absence of light. The deacetylation degree (%DD) was determined with the following equations:

$$\frac{A_{1320}}{A_{1420}} = 0.3822 + 0.03133 * AD \tag{1}$$

$$\% \, DD = 100 - AD \tag{2}$$

where $A_{1320}$ was the absorbance at 1320 nm, $A_{1420}$ was the absorbance at 1420 nm, 0.3822 and 0.03133 are constants reported by Colina et al. [23], AD was the acetylation degree and DD was the deacetylation degree (%).

### 2.7. Chitosan Ash

The official method 942.05 of the AOAC [24] was employed to determine ash content (%A). Two grams of chitosan were heated in a muffle furnace preheated at 550 °C for 4 h. The ash content (% A) was calculated by the following equation:

$$\%A \ = \ \frac{(\text{weight of residue, g})}{(\text{sample weight, g})} * 100 \tag{3}$$

### 2.8. Process Yield

The process yield (% Y) was obtained according to the methodology reported by Hernández Cocoletzi et al. [25]. The results were calculated using the following expression:

$$\%Y \ = \ (\frac{\text{chitosan weight, g}}{\text{Shrimp waste in wet base, g}}) * 100 \tag{4}$$

### 2.9. Molecular Weight (Mw) of Chitosan

This determination was made according to the methodology of Sedaghat et al. [8]. Four concentrations of chitosan in a range of 0.33–1% were employed for measuring the viscosity in a capillary viscometer (Cannon Fenske Routine 150, State College, PA, USA) with a size of 0.4 mm in a water bath at 25 °C. The average molecular weight was calculated by measuring the intrinsic viscosity according to the Mark–Houwink–Sakurada (MHS) equation (Equation (5)).

$$[\eta] = \text{K}(M_w)^a \tag{5}$$

where K and a were the constants, K = $1.81 \times 10^3$ and a = 0.93, and $[\eta]$ was the intrinsic viscosity.

### 2.10. Chitosan Optimization

A rotatable central composite design with two variables with 13 treatments was used for optimization. The fermentation time ($X_1$, h) and total soluble solids ($X_2$, °Brix) were the process variables (Table 1). A quadratic polynomial regression model was assumed for predicting (Y) response variables. Models of the following form were developed to describe the response surfaces (Y):

$$Y \ = \ \beta_0 + \beta_1 X_1 + \beta_2 X_2 + \beta_{12} \ X_1 X_2 + \beta_{11} X_1^2 + \beta_{22} \ X_2^2 \tag{6}$$

where Y is the value of the considered experimental predicted response variable (deacetylation degree of chitosan, chitosan ash or yield process), $\beta_0$ is the constant value, $\beta_1$ and $\beta_2$ are linear coefficients, $\beta_{12}$ is the interaction coefficient and $\beta_{11}$ and $\beta_{22}$ are the quadratic coefficients. For applying the stepwise regression procedure, non-significant terms ($p > 0.05$) were deleted from the second-order polynomial, and a new polynomial was recalculated to obtain a predictive model for each variable [26]. All the results were analyzed by the statistical software "Design Expert" (Version 7.0.0, Stat-Ease Inc., Minneapolis, MN, USA) to determine the optimum conditions for the fermentation process. The optimal levels of these variables were obtained by solving the regression equations and analyzing the response surface contour plots using the same software [4].

### 2.11. Fruit Processing

A total of 25 papaya fruits, cv Maradol, were obtained from a local based on the uniformity of skin color, the absence of physical damage and an average weight of 1200 ± 300 g. The fruits were washed with a 0.02% (*v/v*) sodium hypochlorite solution before being peeled, and the seeds were removed. The fruits were transversely cut to form slices of 1 cm thickness. The slices were randomly divided into 3 lots. One lot was dipped in purified water, another one was dipped in a commercial chitosan solution (1%, *w/v*) and a third one was dipped in an optimized chitosan solution (1%, *w/v*) for 3 min at 5 °C. After

the immersions, 5 slices were placed into 0.25 L polypropylene trays (Nutrigo S.A. de C.V., Saltillo, CO, México) (about 200 g per tray), and the trays were sealed. Forty-five trays per treatment were refrigerated at 5 °C for 10 days. Six trays of each treatment were randomly removed at 2 days intervals for physical and chemical analysis, and 3 trays were removed every 5 days for microbiological analysis. Three replicates per treatment were performed [13].

### 2.12. Physical Analysis of Fresh-Cut Papaya

Color measurements were performed at the center of each slice according to Ayón-Reyna et al. [13] using a Minolta colorimeter (model CR-200; Minolta Co. Ltd., Osaka, Japan) based on the CIELAB color parameters; L∗ and b∗ were analyzed for all the treated slices. Nine measurements were performed per treatment.

Firmness was analyzed using a digital penetrometer (Chatillon DFE 100; AMETEK Inc, Largo, FL, USA) fitted with an 11 mm diameter probe [13]. The pericarp at the center of each slice was penetrated (5 mm depth) at a speed of 50 mm/min. Five measurements were performed per treatment, and the results were expressed in Newtons (N).

### 2.13. Chemical Analysis of Fresh-Cut Papaya

Total soluble solids were determined following the official method 22.014 of the AOAC [24]. The samples were analyzed using a digital refractometer (Hanna instruments, HI 96801, Rhode Island, USA). The results were expressed in °Brix. Twelve slices per replicate were evaluated ($n$ = 12).

The titratable acidity and pH were evaluated following the official methods 942.15 and 981.12 of the AOAC [24], respectively. Twenty g of sample were homogenized with 100 mL neutral distilled water using an Ultra-Turrax homogenizer (IKA T18 basic Ultra-Turrax, Wilmington, NC, USA), and then filtered. The pH of the homogenized solution was measured with a potentiometer (Orion Research Inc., Beverly, MA, USA). Titratable acidity was determined by titration of the homogenized solution with 0.1N NaOH (to a pH value of 8.1 ± 0.2). Five measurements were performed per treatment and the results were expressed as the percentage of citric acid and pH, respectively.

### 2.14. Microbiogical Analysis of Fresh-Cut Papaya

The samples were prepared and evaluated according to the methodology reported by Ayón-Reyna et al. [13]. Two serial dilutions were prepared with papaya samples. In the 1st dilution, 50 g of pulp was mixed with 450 mL peptone water (1%) and homogenized for 1 min under sterile conditions to obtain a 1:10 dilution. Next, this dilution (1 mL) was mixed with 9 mL of BHI broth (brain heart infusion agar; Bioxon BD, Sparks, NV, USA) to obtain the 2nd dilution at 1:100. For the account of mesophilic and psychrophilic microorganisms, molds and yeasts, one hundred microliters of prepared dilutions were inoculated on plate count agar (PCA, Difco BD, Sparks, NV, USA). For the mesophilic microorganisms count, the samples were incubated at 37 °C for 24 to 48 h, and the samples used for the psychrophilic microorganisms count were incubated at 5 °C for 13 to 15 days. To determine molds and yeasts, 100 µL of prepared dilutions were inoculated on plates with Sabouraud agar at 25 °C for 3 to 5 days. The results are expressed as colony-forming units per gram (CFU/g), and 6 measurements were performed per treatment.

### 2.15. Statistical Analysis of Fresh-Cut Papaya

A completely randomized experimental design with 3 replicates was performed, where 3 trays with 5 slices each constituted a replicate. Statistical analysis of the data was performed through factorial variance analysis using Statgraphics Plus 5.1, and the means were compared using Fisher's least significant difference (LSD) test ($p \leq 0.05$).

## 3. Results

### 3.1. Predictive Models

Three predictive models were obtained from fitting the second-order polynomial of Equation (6) for three response variables (deacetylation degree, ash and process yield) as a function of two fermentation process variables (fermentation time and total soluble solids). The experimental results of each response at different combinations of the fermentation process variables are shown in Table 1. The deacetylation degree of chitosan varied from 76.1% to 90.62%, chitosan ash varied from 0.187% to 0.612% and process yield varied from 1.611% to 2.024% (Table 1). These predictive models were tested for adequacy and fitness by the analyses of variance (ANOVA, Table 2).

**Table 2.** Regression coefficients and variance analysis of polynomial models for lactic fermentation extraction of chitosan.

| Regression Coefficients | Deacetylation Degree (%) | | Ash (%) | | Process Yield (%) | |
|---|---|---|---|---|---|---|
| | Coded | Uncoded | Coded | Uncoded | Coded | Uncoded |
| Intercept | | | | | | |
| $\beta_0$ | 85.82 | 89.08 | 0.33 | 0.562 | 1.65 | 2.781 |
| Lineal | | | | | | |
| $\beta_{10}$ | −3.43 | −0.065 | −0.048 | $-4.86 \times 10^{-3}$ | −0.089 | $-3.74 \times 10^{-3}$ |
| $\beta_2$ | −1.63 | 0.565 | 0.027 | 0.014 | 0.014 | 0.063 |
| Quadratic | | | | | | |
| $\beta_{11}$ | −1.69 | $-1.72 \times 10^{-4}$ | 0.096 | $9.74 \times 10^{-6}$ | 0.085 | $8.63 \times 10^{-6}$ |
| $\beta_{22}$ | −3.32 | −0.027 | −0.051 | $4.23 \times 10^{-4}$ | 0.16 | $1.35 \times 10^{-3}$ |
| Interaction | | | | | | |
| $\beta_{12}$ | 3.98 | $3.65 \times 10^{-3}$ | 0.053 | $4.83 \times 10^{-5}$ | | |
| $R^2$ | 0.983 | 0.983 | 0.962 | 0.962 | 0.977 | 0.977 |
| Adjusted $R^2$ | 0.9714 | 0.9714 | 0.9363 | 0.9363 | 0.9666 | 0.9666 |
| Lack of fit (*p* value) | 0.078 | 0.078 | 0.2201 | 0.2201 | 0.2161 | 0.2161 |
| CV (%) | 0.97 | 0.97 | 7.48 | 7.48 | 1.56 | 1.56 |
| Model *p* value | 0.0001 | 0.0001 | 0.0001 | 0.0001 | 0.0001 | 0.0001 |

#### 3.1.1. Deacetylation Degree of Chitosan

According to the analysis of variance (Table 2), the deacetylation degree of chitosan was significantly dependent on the linear terms of fermentation time ($X_1$) (FT, $p < 0.01$) and total soluble solids ($X_2$) (TSS, $p < 0.05$), quadratic terms of FT and TSS ((FT)$^2$, $p < 0.01$, (TSS)$^2$, $p < 0.05$) and FT-TSS interaction ($X_1 X_2$) (FT-TSS, $p < 0.05$. The predictive models for the deacetylation degree were:

Using coded variables,

$$Y_{DD} = 85.82 - 3.43X_1 - 1.63X_2 + 3.98X_1X_2 - 1.69X_1{}^2 - 3.32X_2{}^2 \tag{7}$$

Using original variables,

$$Y_{DD} = 89.079 - 0.065TF + 0.565TSS + 3.65 \times 10^{-3}TF*TSS - 1.727 \times 10^{-4}TF^2 - 0.027TSS^2 \tag{8}$$

The predictive model explained 98.3% of the total variation ($p < 0.05$) in the deacetylation degree values (Table 2), and the lack of fit was not significant ($p > 0.05$). Furthermore, the relative dispersion of the experimental points from the predictions of the models (CV) was found to be 0.97%. According to Milán-Carrillo et al. [26], a good predictive model should have an adjusted $R^2$ (coefficient of determination) $\geq 0.80$, a significance level of $p < 0.05$, coefficients of variance (CV) values $\leq 10\%$ and a lack of fit test >0.05; all these parameters could be used to decide the satisfaction of the modeling. Therefore, our values indicated that the experimental model for the deacetylation degree of chitosan was ade-

quate and reproducible. Maximum values (desirable values) of the deacetylation degree were observed at FT = 25–75h and TSS = 8.44–10% (Figure 1).

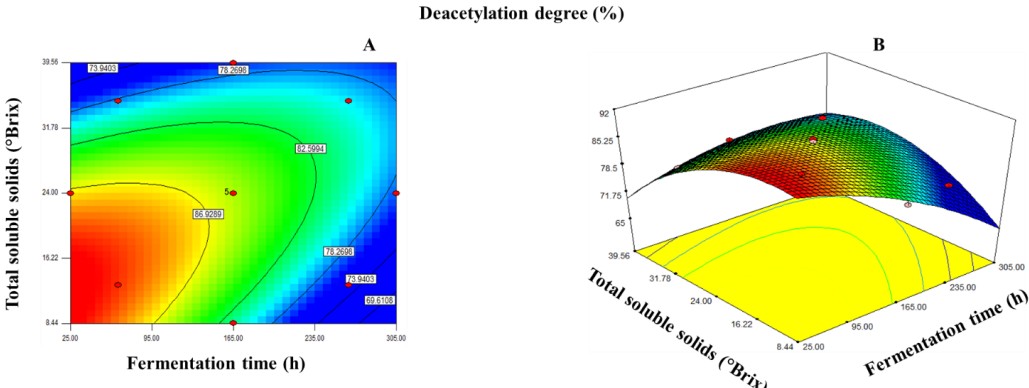

**Figure 1.** Contour plots and response surface (**A**,**B**) for the effect of fermentation time (h) and total soluble solids (°Brix) on the deacetylation degree of chitosan.

### 3.1.2. Chitosan Ash

The chitosan ash model was significantly dependent on linear terms of fermentation time ($X_1$) (FT, $p < 0.01$) and total soluble solids ($X_2$) (TSS, $p < 0.05$)), quadratic terms of FT ($X_1^2$) and TSS ($X_2^2$) ((FT)$^2$, $p < 0.01$; (TSS)$^2$, $p < 0.05$) and FT-TSS interaction ($X_1X_2$) (FT-TSS, $p < 0.05$). The predictive models for the chitosan ash were:

Using coded variables,

$$Y_A = 0.33 - 0.048X_1 + 0.027X_2 + 0.053X_1X_2 + 0.096X_1^2 - 0.051X_2^2 \tag{9}$$

Using original variables,

$$Y_A = 0.62 - 4.86 \times 10^{-3}FT + 0.014TSS + 4.83 \times 10^{-5}FT * TSS + 9.74 \times 10^{-6}FT^2 - 4.23 \times 10^{-4}TSS^2 \tag{10}$$

The predictive model explained 96.2% of the total variation ($p < 0.05$) in the chitosan ash values (Table 2), and the lack of fit was not significant ($p > 0.05$). Furthermore, the relative dispersion of the experimental points from the predictions of the models (CV) was 7.48%. Desirable minimum values of chitosan ash were observed at FT = 165–305h and TSS = 8.44–13% (Figure 2).

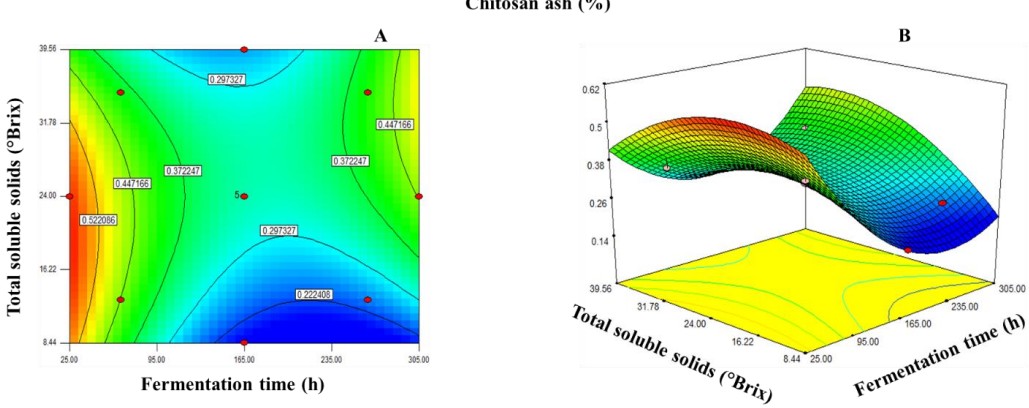

**Figure 2.** Contour plots and response surface (**A**,**B**) for the effect of fermentation time (h) and total soluble solids (°Brix) on chitosan ash content.

### 3.1.3. Process Yield

The process yield model was significantly dependent on linear terms of fermentation time ($X_1$) and total soluble solids ($X_2$) (FT, $p < 0.01$; TSS, $p < 0.01$), as well as the quadratic terms of FT ($X_1^2$) and TSS ($X_2^2$) ((FT)$^2$, $p < 0.01$; (TSS)$^2$, $p < 0.05$). The predictive models for process yield were:

Using coded variables,

$$Y_Y = 1.65 - 0.89X_1 + 0.085X_1 + 0.14X_2 + 0.085X_1^2 + 0.16X_2^2 \qquad (11)$$

Using original variables,

$$Y_Y = 2.781 - 3.74 \times 10^{-3} FT - 0.063 TSS + 8.63 \times 10^{-6} FT^2 + 1.35 \times 10^{-3} TSS^2 \qquad (12)$$

The predictive model explained 97.7% of the total variation ($p < 0.05$) in the process yield values (Table 2), and the lack of fit was not significant ($p > 0.05$). Furthermore, the relative dispersion of the experimental points from the predictions of the models (CV) was 1.56%. Desirable maximum yield values were observed at FT = 25−75h, TSS = 8.44–13% and 35–39.5% (Figure 3).

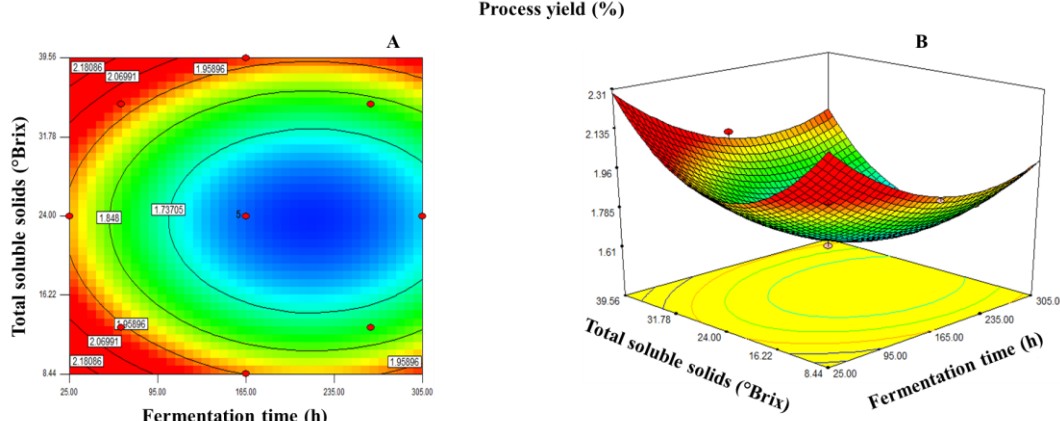

**Figure 3.** Contour plots and response surface (**A**,**B**) for the effect of fermentation time (h) and total soluble solids (°Brix) on process yield of chitosan.

### 3.2. Chitosan Optimization

A graphical method was employed to obtain the optimal conditions of the fermentation process for chitosan extraction. Figures 1A, 2A and 3A (contour plots) show the effects of the fermentation time (FT) and total soluble solids (TSS) on the deacetylation degree, ash and yield of chitosan, respectively. Then, the superposition of these contour plots was carried out to obtain a new contour plot (Figure 4), which was utilized for the observation and selection of the best combination of fermentation process variables for producing optimized chitosan with a maximum deacetylation degree, process yield and minimum levels of chitosan ash. The central point of the optimization region in Figure 4 corresponds to a combination of the process variables of FT = 108h and TSS = 8.74%. Under these conditions, chitosan was estimated with a deacetylation degree of 86.3%, a process yield of 2% and an ash content of 0.29%. To validate the aforementioned optimal conditions, chitosan was obtained under the best combination of lactic fermentation (fermentation time/total soluble solids) was produced in quintuplicates; the experimental chitosan had a deacetylation degree of 83%, an chitosan ash content of 0.23%, a process yield of 2.03% and a molecular weight of 107.5 kDa.

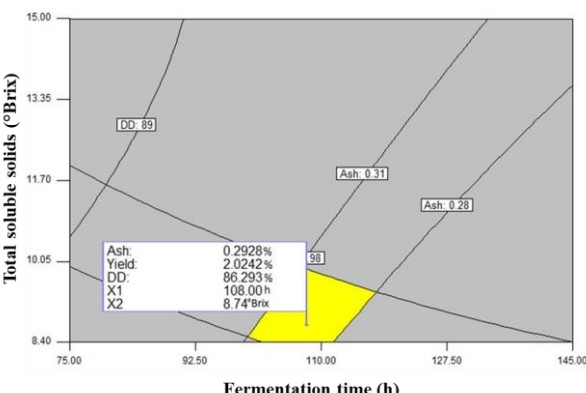

**Figure 4.** Region of the best combination of lactic fermentation (fermentation time/total soluble solids) for optimized chitosan production.

### 3.3. Physical Analysis of Frush-Cut Papaya

During fresh-cut processing, metabolic fruit changes are presented as a consequence of peeling and cutting steps; also, enzymatic browning decreases the tissue luminosity (L*) during storage [15,17,18]. In this work, no changes in the L* value were noticed between the treated fruits at the beginning of their storage. Nevertheless, at the end of the refrigerated period, the control treatment had the highest loss of L∗ value, with 12.6%. On the other hand, the chitosan (commercial and optimized) treatments only showed 8% loss, and we did not observe a significant difference between the chitosan treatments (Figure 5A). The b* value in the CIELab system corresponds to the yellow color, and a decrease could be related to enzymatic browning [18,27]. The initial average value of the b* parameter was 36, and it gradually decreased during storage for all the treatments (Figure 5B). On day 10, the control treatment had the lowest b∗ value, followed by the chitosan (commercial and optimized) treatments, which did not show statistical difference between them but with the control.

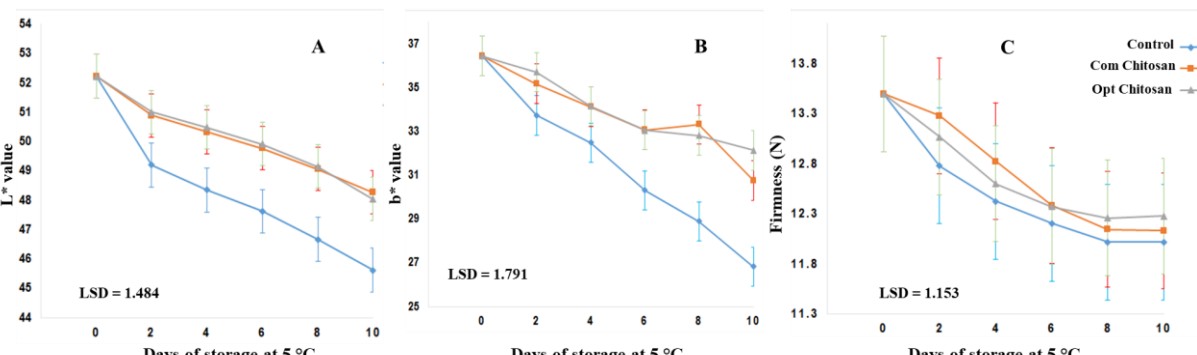

**Figure 5.** Color changes (L* (**A**) and b* (**B**) values) and firmness (**C**) during storage at 5 °C of papaya slices previously dipped in a chitosan solution. Control = fresh-cut papaya slices without application of chitosan film; Quit Com = fresh-cut papaya slices with application of commercial chitosan film; Quit Opt = fresh-cut papaya slices with application of chitosan film obtained under optimized lactic fermentation conditions. Each point represents the mean of 15 replicates. Vertical bars indicate LSD ($p \leq 0.05$).

Fruit firmness is an important attribute that dictates the post-harvest life and quality of the fruit due to it is relationship with water content and metabolic changes [16,28]. At the processing time, the fruits showed firmness values close to 13.5 N, which decreased during the second day and continued a steady decline throughout storage (Figure 5C). No statistically significant differences ($p > 0.05$) were found between the treatments in the storage days studied. At the end of their storage, the firmness showed values close to 12.1 N.

### 3.4. Chemical Analysis of Fresh-Cut Papaya

During the storage period, an increase in the pH values was observed for all the treatments, starting from a value of 5.4 at the beginning (Figure 6A). This increase could be attributed to the degradation of organic acids used as substrates in various biochemical processes [13]. Papaya slices untreated with chitosan presented the highest pH values throughout the storage, showing a statistically significant difference ($p < 0.05$) with respect to the treatments with chitosan (commercial and optimized), which presented the lowest pH values during storage. In addition, the papaya slices treated with any chitosan did not show significant statistical differences between them, which could mean that both treatments have similar barrier properties.

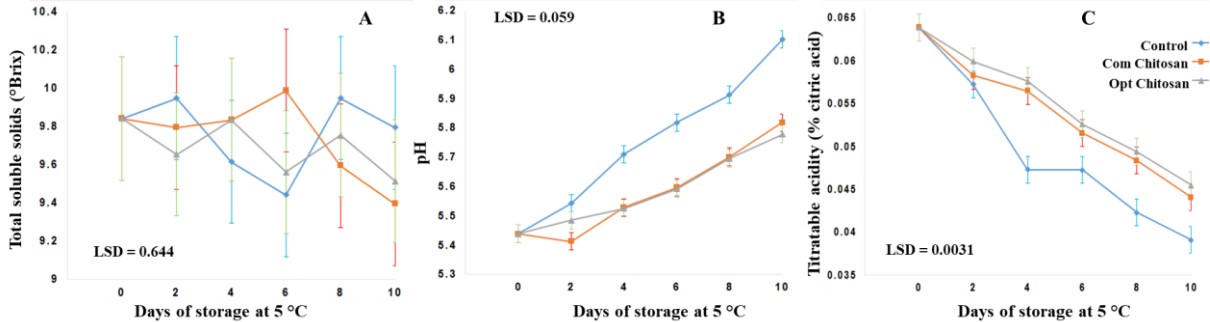

**Figure 6.** Changes in pH (**A**), total soluble solids (**B**) and titratable acidity (**C**) during storage at 5 °C of papaya slices previously dipped in a chitosan solution. Control = fresh-cut papaya slices without application of chitosan film; Quit Com = fresh-cut papaya slices with application of commercial chitosan film; Quit Opt = fresh-cut papaya slices with application of chitosan film obtained under optimized lactic fermentation conditions. Each point represents the mean of 15 replicates. Vertical bars indicate LSD ($p \leq 0.05$).

Papaya slices showed an initial TSS content of about 9.84 °Brix (Figure 6B), which is comparable with values reported by Ali et al. [29] and Ayón-Reyna et al. [13] in the whole fruits and the slices of papaya, respectively. However, the TSS content did not show a stable behavior during storage without showing significant differences ($p > 0.05$) between the treatments.

At the initial time of processing, the titratable acidity was 0.064%. Figure 6C shows the corresponding values expressed as a percentage of citric acid for the different treatments during the storage of papaya slices at 5 °C. A decrease in this parameter can be observed in all the treatments from day two until the end of storage (day 10). The control fruit had the lowest percentage of citric acid with statistically significant differences ($p < 0.05$) with those of the chitosan (commercial and optimized)-treated fruits. These results can be related to the pH values because the decrease in titratable acidity is linked with a pH increase.

### 3.5. Microbial Analysis of Fresh-Cut Papaya

In minimally processed products, the structure of the fruit is damaged by cutting, which causes nutrient output, making it more susceptible to the growth of microorganisms. Additionally, the minimally processed fruits are stored at low temperatures (4–6 °C) to increase the shelf life [13,16–18]. The results of the different treatments and microorganisms analyzed in minimally processed papaya during the storage period are presented in Figure 7. An increase in the number of microorganisms presence (bacteria, yeast and molds) could be observed for all the treatments. The chitosan (commercial and optimized) treatments showed a higher inhibition of the growth of microorganisms than the control did at days 5 and 10 of storage. Additionally, the chitosan treatments showed similar behavior throughout the storage period, without statistical differences ($p > 0.05$) between them. At the end of the storage for mesophilic bacteria, the control treatment had a value of $2.45 \times 10^6$ CFU/g, and the lowest bacteria counts were submitted to the treatments

containing chitosan (commercial and optimized), with values of $9.5$–$9.53 \times 10^5$ CFU/g (Figure 7A). The chitosan treatments (commercial and optimized) reduced the proliferation of mesophilic bacteria five-fold in comparison with that of the control treatment. Likewise, the psychrophilic bacteria count at the end of the storage for control treatment was $3.19 \times 10^5$ CFU/g, while those of the commercial chitosan and optimized chitosan treatments were $1.39 \times 10^5$ and $1.51 \times 10^5$ CFU/g, respectively (Figure 7B). On the other hand, at the end of the storage the control treatment had a fungi and yeast count of $1.58 \times 10^6$ CFU/g. In contrast, chitosan (commercial and optimized) treatments had fungi and yeast growth values of $2.28 \times 10^5$ and $1.98 \times 10^5$, respectively (Figure 7C).

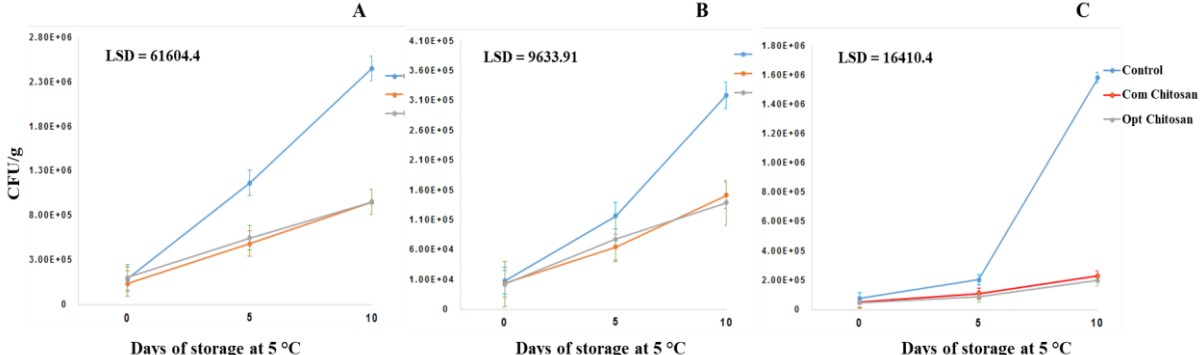

**Figure 7.** Counting of mesophilic (**A**), psychrophilic (**B**), molds and yeasts (**C**) microorganisms during storage at 5 °C of papaya slices that had been previously dipped in a chitosan solution. Control = fresh-cut papaya slices without application of chitosan film; Quit Com = fresh-cut papaya slices with application of commercial chitosan film; Quit Opt = fresh-cut papaya slices with application of chitosan film obtained under optimized lactic fermentation conditions. Each point represents the mean of 9 repetitions. Vertical bars indicate LSD ($p \leq 0.05$).

## 4. Discussion

Chitosan obtained from chitin with a deacetylation degree of 70% or more is considered to be a good final product and could be related to a high-potential application [30,31]. In our study, chitosan obtained under the best processing conditions has a deacetylation degree of 83%; therefore, optimized chitosan could be used in the food industry. In this sense, optimized chitosan has a greater deacetylation degree than the values reported by Sierra et al. [32], Zhang et al. [33] and Aneesh et al. [7]. Additionally, optimized chitosan has a similar deacetylation degree to that reported by Tokatli and Demirdoven [21]; however, it has a lower degree than the value reported by Sedaghat et al. [8] and commercial chitosan (87%). The differences with our results and those of other authors could be attributed to the process conditions and shrimp species [20,21].

Applications of chitosan have been related to the absence of proteins and minerals; therefore, deproteination and demineralization steps are important for chitosan production [30,34]. Likewise, it has been reported that a suitable chitosan extraction for food industry applications must have an ash content lower than 1% because minerals influence in the solubility and viscosity of chitosan [21,35]. In this sense, optimized chitosan has an ash content of 0.23%, which is lower than those reported by Islam et al. [36], Czechowska-Biskup et al. [37], Neves et al. [34] and commercial chitosan. The lower ash content in comparison to that of chitosan obtained by chemical extraction could be attributed to the lactic fermentation process used, which transforms the calcium carbonate of shrimp waste into calcium lactate that dissolves in the liquor; also, the fermented solid was subsequently subjected to a demineralization stage with a solution of hydrochloric acid at a low concentration (solution 1N, ratio 1/10 (*w/v*), period = 1 h) to obtain the chitosan [8,20].

The chitosan obtained in the best processing conditions has a lower process yield compared with those reported by Parada et al. [38], Islam et al. [36], Sedaghat et al. [8] and Tokatli and Demirdoven [21]. However, optimized chitosan has a similar process yield as

those reported by Badawy and Rabea [39] and Gamal et al. [4]. In this work, a relationship between the ash content and yield was observed; at a higher fermentation time, a lower ash content was obtained, and therefore, a lower chitosan yield was obtained. The process yield of chitosan could be affected by the process conditions due to the deproteination, demineralization and deacetylation steps, which separate the compounds from chitin to obtain chitosan; also, during lactic fermentation, the proteins and minerals were transformed into hydrolyzed proteins and calcium lactate by acidic medium, which can change the polymeric chain and affect the process yield [7,20,21].

Optimized chitosan has a lower molecular weight than those reported by Parada et al. [38], Islam et al. [36], Sedaghat et al. [8] and Tokatli and Demirdoven [21], also, optimized chitosan has a similar molecular weight to those reported by Teli and Sheikh [40] and Zhang et al. [33]. Differences in the molecular weight could be attributed to the processing conditions, as lactic fermentation can separate the proteins, minerals and other compounds from chitin that can be found in the liquor. At the same time, the fermented solid was subsequently treated with chemicals to obtain chitin and chitosan with sodium hydroxide that degraded the polymeric chain [20,21,41,42]. On the other hand, a relationship between the deacetylation degree and molecular weight of chitosan with biological characteristics has been reported due to a high deacetylation degree and a low molecular weight, which could enhance chitosan solubility and increase its potential applications [7,31].

During the fresh cutting of fruits, metabolic changes are presented in the cutting and peeling steps, and enzymatic browning decreases the luminosity (L*) value during the days of storage [15,17,18]. In this sense, the optimized chitosan treatment inhibited the decrease in the L* and b* values without statistical differences with commercial chitosan; these results are in concord with those reported by several authors [13,15,17,27] for chitosan coatings on fresh-cut fruits. However, these results are not in agreement with reports of Nascimento et al. [18]. The decrease in the L* and b* values of the papaya slices could be attributed to the loss of moisture by overripe and texture alteration, showing enzymatic browning [13,27]. Additionally, the chitosan treatments showed smaller L* and b* losses because the chitosan coating may act as a barrier, preventing enzymatic browning and color changes, and mitigating the ripening process [17,18].

Fruit firmness is an important attribute of post-harvesting due to it is related to water content and metabolic changes [16,28]. All the treatments (control, optimized and commercial chitosan) showed a firmness decrease without statistical difference during the days of storage. These results were similar to those reported by several authors [13,29,43,44] for chitosan coatings on minimally processed fruits. However, these results are not in agreement with those reported by Chiabrando and Giacalone [15], Hesami et al. [45], Khalil et al. [46], Shyu et al. [17], Vivek and Subbarao [28] and Wang et al. [47]. The firmness reduction could be attributed to different factors, such as natural ripening, cutting and peeling processes, which cause cell deterioration, higher pectinolytic enzyme activity and a higher respiration rate [13,28,29]. In this sense, different strategies have been studied for maintaining the firmness of fresh-cut fruits, such as high chitosan concentration coatings at low-temperature storage and hydrothermal-calcium chloride treatments because combined treatments could form a modified atmosphere that reduces the water loss and respiration rate, and also, could inhibit the metabolic processes (such as polygalacturonase activity) during senescence [13,16,17,28,44,45].

Total soluble solids (TSS) are considered to be an important indicator of the sugar content in fruits such as papaya [11,28]. Optimized chitosan did not have a significant effect on the other treatment throughout their storage. Similar results were reported by Argañosa et al. [48], Ayón-Reyna et al. [13] and Shyu et al. [17] who observed that the content of TSS in fresh-cut fruits such as papaya was not affected by the storage time and the fruit preparation. According to these results, Sañudo-Barajas et al. [49] and Ali et al. [29] reported that there is no accumulation of sugars in the papayas (cultivar Maradol) after the treatments that accelerate ripening degradation of the cell wall and an increase in the enzymatic activity.

pH and titratable acidity are quality parameters for fruits [28] because the organic acids have an important role in its flavor [11]. Chitosan treatments (optimized and commercial) have a lower increase in pH and a lower decrease in titratable acidity ($p < 0.05$) than those of the control treatment during the days of storage. Similar results were obtained by Gonzalez-Aguilar et al. [27], Ali et al. [43], Chávez-Sánchez et al. [19], Ayón-Reyna et al. [13], Chiabrando and Giacalone [15], Vivek and Subbarao [28] and Khalil et al. [46] however, these results are not in agreement with those reported by Shyu et al. [17] and Nascimento et al. [18] for fresh-cut fruits. Chitosan coatings on fresh-cut fruits have been reported to have decreased fruit metabolism because chitosan films form a modified atmosphere that produces regulating gas exchange and a slow respiration rate, requiring fewer organic acids [11,13,28,44,45].

Microbial safety is an important parameter that needs to be studied for the preservation of minimally processed fruits [15]. In these products, the structure of the fruit is damaged by cutting, which produces nutrient output, making it more susceptible to the growth of microorganisms [13,16–18]. The optimized chitosan coating inhibited the microorganism growth, with a statistical difference with the control treatment. These results are in concordance with those reported by Ayón-Reyna et al. [13], Dotto et al. [11], Chiabrando and Giacalone [15], Gurjar et al. [16], Shyu et al. [17] and Vivek and Subbarao [28] for the microbiological quality of fresh-cut fruits. Different mechanisms for the microbial inhibition of chitosan coatings have been proposed; in this sense, some authors reported that chitosan coatings could modify the atmosphere, which generates selective gas permeability, inhibiting spoilage bacteria by inactivating enzymes and causing nutrient competition. In addition, the cationic nature of chitosan could affect the permeability and integrity of microorganisms throughout intracellular transport by the interaction between the amino groups of chitosan and the electron interaction of microbial cell membranes [13,16,17,50]. Therefore, optimized chitosan obtained by combining biological and chemical methods could maintain the quality and inhibit microbial growth in processed fruits.

## 5. Conclusions

The optimized fermentation process (fermentation time of 108 h and total soluble solids of 8.74 °Brix) produced chitosan with good characteristics and properties, which was similar to commercial chitosan, but with less frequent use of polluting chemical agents. Optimized chitosan inhibited microbial growth and the loss of color and acidity; also, it increased the pH of fresh-cut papaya slices, which maintained their quality characteristics during the storage time. Chitosan obtained from the optimized lactic acid fermentation chemical treatment of shrimp by-products could be applied as a coating to increase the shelf life of different fresh-cut fruits.

**Author Contributions:** Conceptualization, L.A.C.-B. and R.G.-D.; Methodology, L.A.C.-B., M.O.V.-G., J.M.-Á. and H.S.L.-M.; Software, L.A.C.-B. and R.I.C.-L.; Validation, R.I.C.-L., J.M.-Á. and R.G.-D.; Formal analysis, L.A.C.-B. and R.G.-D.; Investigation, L.A.C.-B. and R.I.C.-L.; Resources, R.G.-D.; Data curation, L.A.C.-B.; Writing—original draft preparation, L.A.C.-B. and R.I.C.-L.; Writing—review and editing, M.O.V.-G., J.M.-Á. and H.S.L.-M.; Visualization, R.I.C.-L. and R.G.-D.; Supervision, J.M.-Á., R.I.C.-L. and R.G.-D.; Project Administration, R.G.-D.; Funding acquisition, R.I.C.-L. and R.G.-D. All authors have read and agreed to the published version of the manuscript.

**Funding:** This research was supported by the "Consejo Nacional de Ciencia y Tecnología (CONA-CYT)" and the "Facultad de Ciencias Químico-Biológicas, Universidad Autónoma de Sinaloa".

**Data Availability Statement:** Not applicable.

**Acknowledgments:** Authors dedicated this work to the memory of our beloved friend, MC Marco Antonio Parra Inzunza. Additionally, the authors are grateful to the Facultad de Ciencias Químico-Biológicas, Universidad Autónoma de Sinaloa and to the Consejo Nacional de Ciencia y Tecnología for financing this research and providing a fellowship to LACB.

**Conflicts of Interest:** The authors declare no conflict of interest.

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
