# Peer review of "Effect of Optimized Chitosan Coating Obtained by Lactic Fermentation Chemical Treatment of Shrimp Waste on the Post-Harvest Behavior of Fresh-Cut Papaya (Carica papaya L.)"

_fermentation, doi:10.3390/fermentation9030220_

Round 1
Reviewer 1 Report
The work done by Luis and co-authors described an optimized less polluting biological-chemical method to get chitosan from shrimp waste, using a successive lactic fermentation and chemical process. The method is practically useful. The produce is in good quality of high deacetylation degree and low ash content. The manuscript is well-written.
Author Response
Dear reviewer, thank you for your comment and time to review this paper.
Reviewer 2 Report
The manuscirpt nends some revisions.
1. The important data of results should be added in the abstract section.
2. The introduction should be extended the content by concluded some related articles.
3. The physical property and activity of chitosan and its coating should be added such as SEM, and antimicrobial activity.
4. The more related articles should be cited to discuss the effect mechanism.
5. The English language should be revised in the whole manuscript.
6. The structure of methods about indexes determination should be improved.
Author Response
- The important data of results should be added in the abstract section.
Response.- Dear reviewer, thank you for your comment. The results were resumed in the abstract section due to the word limit (200 words). Please check the lines 23-37.
- The introduction should be extended the content by concluded some related articles.
Response.- Dear reviewer, thank you for your comment. We added information about the fruit and chitosan coating properties in the introduction section. Please check the lines 60-73.
- The physical property and activity of chitosan and its coating should be added such as SEM, and antimicrobial activity.
Response.- Dear reviewer, we appreciate your comment. The chitosan coating properties for maintaining the quality of fresh-cut fruits are in the lines 60-66. Also, the effect of chitosan coating on papaya quality parameters and microorganism growth were discussed in the lines 483-485, 519-522, and 530-537.
- The more related articles should be cited to discuss the effect mechanism.
Response.- Dear reviewer, thank you for your comment. The discussion of the chitosan coating effect on papaya quality parameters and microorganism growth are in the lines 483-485, 519-522, and 530-537.
- The English language should be revised in the whole manuscript.
Response.- Dear reviewer, we appreciate your comment. We reviewed the English in the whole manuscript.
- The structure of methods about indexes determination should be improved.
Response.- Dear reviewer, thank you for your comment. We reviewed all the equations, and we described all the parameters used.
Reviewer 3 Report
The manuscript consists of preparing chitosan coating, optimization, and postharvest application on papaya fruit slices. The following changes need to be in the manuscript.
Title: It should be short
Introduction: Need to add a paragraph about the importance of papaya, and highlight the main problem papaya industry and the effect of chitosan on papaya.
Line 60-61: The word “poorly” is not proper, change to hardly or rephrase the sentence.
Material and methods:
Line 75: Shrimp heads preparation was realized according… change to… Shrimp heads preparation was performed according
Line 78: 3785 mL, convert into Litter
Line 83-84: rewrite, shrimp waste is used twice in this sentence.
Q1: Why you selected Papaya for evaluation of the effect of Chitosan?
Q2: For how much time you dipped the papaya in water and Chitosan coating?
Results: A lot of basic quality parameters are missed to evaluate the effect of Chitosen on Papaya fruit quality. I recommend you should add these parameters; Weight loss, Total antioxidant, Total anthocyanin, and Total flavonoids.
Author Response
Title: It should be short
Response.- Dear reviewer, thank you for your comment. The title was shorted. Please check the lines 1-3.
Introduction: Need to add a paragraph about the importance of papaya, and highlight the main problem papaya industry and the effect of chitosan on papaya.
Response.- Dear reviewer, we appreciate your comment. A paragraph was added to the manuscript. Please check the lines 60-73.
Line 60-61: The word “poorly” is not proper, change to hardly or rephrase the sentence.
Response.- Dear reviewer, thank you for your comment. The paragraph was changed. Please check the lines 74-77.
Material and methods:
Line 75: Shrimp heads preparation was realized according… change to… Shrimp heads preparation was performed according
Response.- Dear reviewer, thank you for your comment. The sentence was changed.
Line 78: 3785 mL, convert into Litter
Response.- Dear reviewer, thank you for your comment. We change the volume unit.
Line 83-84: rewrite, shrimp waste is used twice in this sentence.
Response.- Dear reviewer, thank you for your comment. We edited the sentence. Please check the line 101.
Q1: Why you selected Papaya for evaluation of the effect of Chitosan?
Response.- Dear reviewer, we appreciate your comment. Papaya is a tropical and temporal fruit with high bioactive compounds; however, its handling is complicated due to its weight and size. Therefore, strategies have been sought to position it in the minimally processed food market. It has been reported that fresh-cut papaya is very susceptible to the growth of microorganisms. Also, our research group has demonstrated the potential of chitosan in maintaining the characteristics of fresh-cut papaya, so the objective of this work is to demonstrate that chitosan obtained under optimal conditions by a sequential method of lactic fermentation-chemical treatment can be used as a coating to inhibit the growth of microorganisms and maintain quality parameters of the fruit. Please check the lines 60-73.
Q2: For how much time you dipped the papaya in water and Chitosan coating?
Response.- Dear reviewer, thank you for your comment. All the treatments were dipped for 3 minutes at 5 °C. Please check the line188.
Results: A lot of basic quality parameters are missed to evaluate the effect of Chitosan on Papaya fruit quality. I recommend you should add these parameters; Weight loss, Total antioxidant, Total anthocyanin, and Total flavonoids.
Response.- Dear reviewer, thank you for your comment. The objective of this work was studying the effect of chitosan coatings obtained from optimized successive lactic acid fermentation-chemical treatment of shrimp by-products on maintain quality parameters and microorganisms’ inhibition. All the methodology used in this work provided us information for this objective. However, in future studies we will carry out analyses for molecular mechanism of chitosan coating, physical and barrier characteristics of chitosan coating, as well as antioxidant capacity and phenolic compounds identification of fresh-cut papaya dipped on optimized chitosan coating.
Reviewer 4 Report
This manuscript is interesting, but the use of chitosan as a coating has been used for both vegetables and fruits. However, obtaining chitosan through the fermentation process is a good use of shrimp waste. It needs a few fixes.
Abstract: L25: should be "postharvest"
Please add in the introduction examples of the use of chitosan coatings in the food industry.
Citations in the text should be in accordance with the requirements of the journal (L81-82; L137; L177; L203; L249; L368; and in discussion section).
You can combine results and discussion.
In conclusions please provide quantitative results.
Author Response
Abstract: L25: should be "postharvest"
Response.- Dear reviewer, thank you for your comment. The word was changed.
Please add in the introduction examples of the use of chitosan coatings in the food industry.
Response.- Dear reviewer, thank you for your comment. We added information on chitosan coating in lines 63-66.
Citations in the text should be in accordance with the requirements of the journal (L81-82; L137; L177; L203; L249; L368; and in discussion section).
Response.- Dear reviewer, we appreciate your comment. All the citations were edited.
You can combine results and discussion.
Response.- Dear reviewer, thank you for your comment. The template for fermentation submission has results and discussion separately. Please check the following link https://www.mdpi.com/journal/fermentation/instructions.
In conclusions, please provide quantitative results.
Response.- Dear reviewer, we appreciate your comment. We added the optimum conditions to chitosan extraction from shrimp by-product in the conclusions. Please check lines 541-542.
Round 2
Reviewer 3 Report
Accepted in Current form